# Fear of Covid 19 during the third wave of infection in Norwegian patients with type 1 diabetes

Grethe Åstrøm Ueland [1,2]*, Tony Ernes[1], Tone Vonheim Madsen[1], Eystein Sverre Husebye[2,3], Sverre Sandberg[1,4,5], Karianne Fjell Løvaas[1], John Graham Cooper[1,6]

1 Norwegian Quality Improvement of Laboratory Examinations (Noklus), Haraldsplass Deaconess Hospital, Bergen, Norway, 2 Department of Medicine, Haukeland University Hospital, Bergen, Norway, 3 Department of Clinical Scinece, University of Bergen, Bergen, Norway, 4 Department of Medical Biochemistry and Pharmacology, Haukeland University Hospital, Bergen, Norway, 5 Department of Global Public Health and Primary Care, Faculty of Medicine, University of Bergen, Bergen, Norway, 6 Department of Medicine, Stavanger University Hospital, Stavanger, Norway

* geas@helse-bergen.no

## Abstract

### Objective

To study the fear of Covid 19 infection among Norwegian patients with type 1 diabetes.

### Research design and methods

Fear of Covid 19 scale, a validated scale assessing the fear of Covid 19, was sent electronically to 16255 patients with type 1 diabetes in May 2021. The items are rated on a scale from 1 to 5 (total scores from 7 to 35). The higher the score, the greater the fear.

### Results

10145 patients, 52% of the Norwegian adult type 1 diabetes population, completed the questionnaire. The mean total fear score was 13.8 (SD 5.8). Women experienced more fear than men (OR 1.96), and fear increased significantly with increasing age for both genders (p<0.05). Fear increased with increasing BMI, more pronounced for men than women. Fear was positively correlated to HbA1c (Spearman rho 0.067, p<0.05), and significantly increased in patients with micro- and macrovascular complications, compared with patients without complications (p<0.05). Smokers showed increased fear compared with non-smokers, (1.59 (1.39–1.81)), and non-European patients reported more fear than Europeans (OR of 2.02 (95% CI 1.55–2.63).

### Conclusion

Assessment of fear of Covid 19 in the type 1 diabetes population in Norway revealed an overall low fear during the third wave of infection. Patients considered to be at high risk of serious disease, such as older individuals, smokers and obese individuals expressed more

**Data Availability Statement:** Sharing of individual participant data with third parties was not specifically included in the informed consent of the

study, and unrestricted distribution of such data may pose a potential threat of revealing participants' identities. To minimise this risk, researchers who wish to inquire about access to individual participant data that underlie the results reported in this article can submit a request to the The Norwegian Adult Diabetes registry (noklus@noklus.no). To gain access, researchers will need to sign a data access agreement and obtain the approval of the local ethics committee.

**Funding:** The author(s) received no specific funding for this work.

**Competing interests:** The authors have declared that no competing interests exist.

fear than low risk individuals. The degree of fear was also associated with sex, ethnicity, educational/working status, glycemic control and presence of complications.

## Introduction

The coronavirus 2019 (Covid 19) is included in the family of β-coronavirus that affects pulmonary gas exchange and triggers a cytokines storm. Fever, dry cough, sneezing, shortness of breath and respiratory distress are the most common symptoms of Covid-19. Inflammation, hyper-coagulation, decreased lymphocytic count with increased neutrophilic count are often observed during the course of the disease. People with a weak immune system are more susceptible to an attack of coronavirus [1].

The Covid 19 pandemic has, to date, infected more than 245 million individuals and caused more than 5 million deaths worldwide [2]. The high infection rate and relatively high mortality rate, as well as limited effective treatment, has led to the Covid 19 pandemic potentially triggering fear and anxiety. Current evidence suggests that a psychiatric epidemic is co-occurring with the Covid 19 pandemic [3]. In line with this, some evidence suggests that infectious disease-related public health emergencies (like pandemics) may increase the suicide risk [4]. The disruption of normal life because of a government-imposed lockdown or quarantine rules has significantly contributed to mental health problems globally [5]. Fear of Covid 19 and an expected increase in psychological problems is an area that merits further investigation during and after the Covid 19 pandemic.

The risk of developing fear and anxiety is probably higher in elderly people, and in people with comorbidities, populations that are shown to have more frequent hospitalization, more severe disease, and higher mortality rates if infected with Covid 19 [6].

In 2016, the prevalence of diagnosed type 1 diabetes was 0.55% in the United States [7]. Diabetes mellitus and other chronic conditions such as severe kidney disease, ongoing cytostatic treatment, heart failure, severe immune deficiency and Downs syndrome carry increased risk of hospitalization for Covid 19 infections, and increased Covid 19 related mortality [6, 8]. A previous study demonstrated that hypertension and diabetes mellitus were the most common chronic illnesses in patients admitted to hospitals with Covid 19 [9].

Whether the increased risk is directly connected to diabetes or to associated factors such as age, obesity or other comorbidities is largely unknown. Regarding differences in severity of Covid 19 infection among subgroups of diabetes mellitus, three British studies indicate that type 1 and type 2 diabetes were both independently associated with significant increased odds of in-hospital death with Covid 19, but more pronounced in type 1 diabetes [10–12]. Furthermore, the French CORONADO study of 1317 diabetes patients hospitalized for Covid 19 infection (mostly type 2 diabetes) found that age, treatment for obstructive sleep apnea, microvascular and macrovascular complications were independently associated with the risk of death on day 7 [13].

In Norway patients with diabetes mellitus are advised to get an annual flu vaccine because of the increased risk of short-term diabetes complications and an increased risk of pneumonia. Despite this, people with diabetes mellitus were not given high priority during the rollout of the Covid 19 vaccination program in Norway. People with diabetes in older age groups have been prioritized, but diabetes as an underlying disease only qualified for the fifth of nine priority groups in the vaccine queue, and then only for patients above the age of 55 years. This may

have contributed to more social isolation for diabetes patients than the background population.

In March 2020, Ahorsu et al published The Fear of Covid 19 Scale (FCV-19S), a seven-item scale, shown to have robust psychometric properties [14]. The scale has been found reliable and valid in assessing fear of Covid 19 among the general population, and has been translated into 35 languages, including Norwegian. We present the results of the fear of Covid 19 score collected electronically from 10145 Norwegian patients with type 1 diabetes who were registered in the Norwegian diabetes register for adults (NDR-A) during the third wave of the pandemic in Norway, in May 2021. The aim of this study is to assess the fear of Covid 19 among Norwegian patients with type 1 diabetes, and correlate the findings to demographic parameters, metabolic control and complications.

## Methods and material

### Participants/data collection

A total of 21484 individuals with type 1 diabetes were registered in the NDR-A in May 2021. Of these, 17828 people with type 1 diabetes aged ≥18 years had attended at least one consultation at a Norwegian diabetic outpatient clinic during the last 15 months. In May 2021, the NDR-A sent the fear of Covid 19 questionnaire electronically via Helsenorge and Digipost to the 16255 (91%) patients who were digitally active and reachable on at least one of the platforms. A reminder was sent after 14 days. In total, 10217 (64%) answered the questionnaires before the deadline at the end of May 2021. Patients who had not answered all seven questions (n = 72) were excluded, leaving 10145 patients to be included in the main calculations.

### Measurements

Clinical and sociodemographic variables like ethnic origin, gender, age, diabetes duration, glycaemic control ($HbA_{1c}$), insulin regimen, long-term diabetes complications and smoking habits were retrieved from the NDR-A. Self-reported data on education and employment status were obtained from supplementary questions included in the PROM questionnaire.

### Fear of Covid 19 scale (FCV-19S)

The FCV-19S is a seven-item scale that assesses the fear of Covid 19. The seven items (e.g. "I am most afraid of corona") are rated on a 5-point scale from 1 (strongly disagree) to 5 (strongly agree) with total scores ranging from 7 to 35. The higher the score, the greater the fear of Covid 19 [14]. The internal consistency reliability for the Norwegian version of the FCV-19S has been found to be very good when assessed by Cronbach's alpha (0.88) [15].

### Statistical analysis

Descriptive statistics were used to quantify patient characteristics. Continuous variables were reported as mean (SD), and median (range). Categorical variables were reported as number (n) and percent (percentage). Logistic regression analysis for binary categorical outcome variable was performed for establishing the association with the demographic indicators, where the outcome variable (Covid fear) was computed as a binary variable using count of any response as "agree" or "strongly agree" for fear questions.

### Ethical considerations

All the participants have given written informed consent that their personal health data can be registered in the NDR-A and used for research. The NDR-A has ethical approval to collect

PROMs in addition to demographic and clinical data from people with diabetes enrolled in the register. The present study has received approval from the Regional Committee for Medical and Health Research Ethics (REK Vest; ref. no. 171685) to extract data from the register for analyses.

## Results

### Study population

A total of 10217 patients were included in the study (see "Fig 1" for the selection of patients), where 10145 completed the FCV-19S, and 72 patients had only partly completed the FCV-19S. Furthermore, 7611 patients did not answer the questionnaire, or were not digitally active and could therefore not be contacted. Table 1 shows demographic characteristics for the patients that answered the form completely, compared with patients that did not answer, or only partially completed the FCV-19S.

Overall, more men (54.1%) than women answered the questionnaire and were included in the study. The majority of the study population were between 30 and 60 years of age (60.6%).

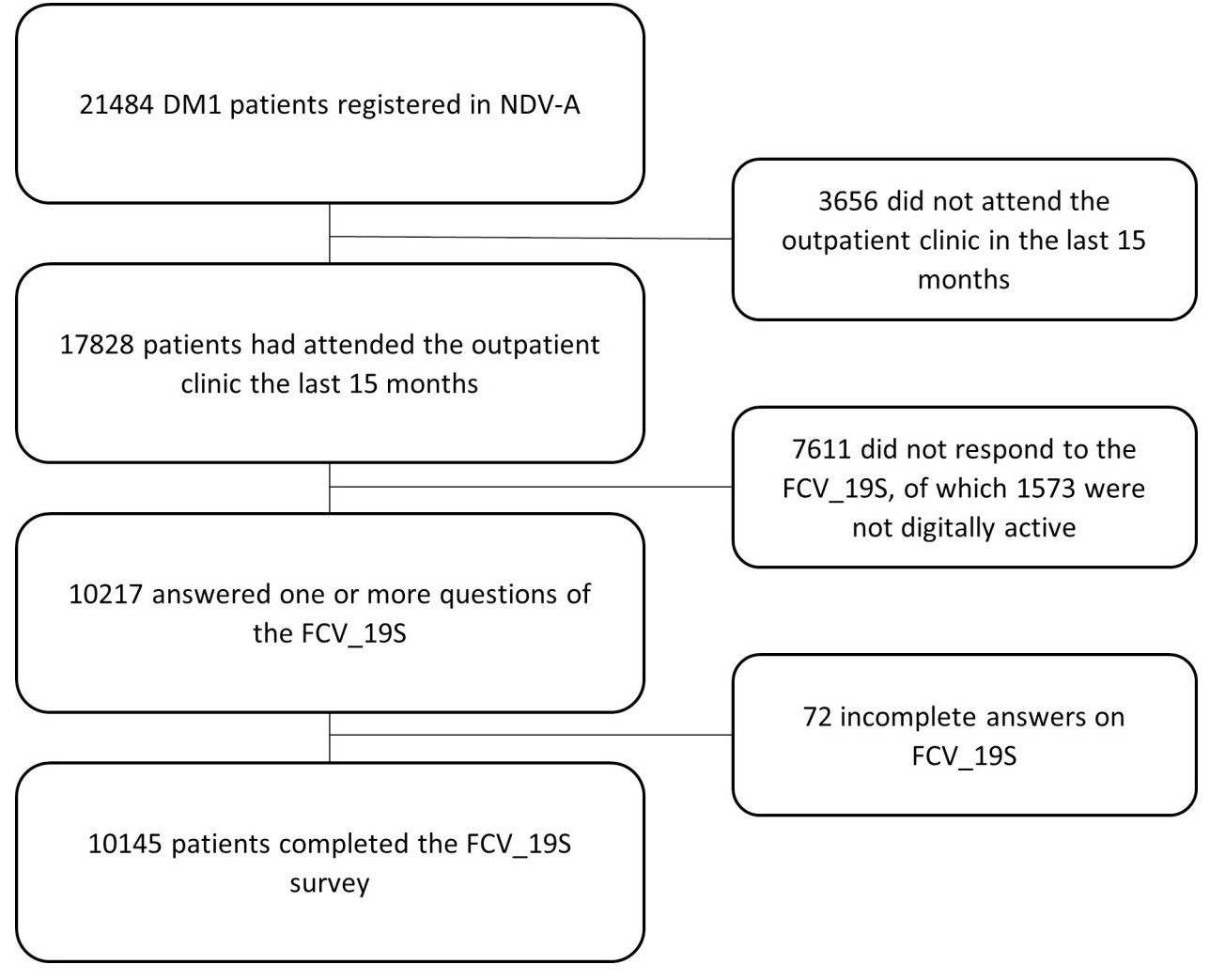

**Fig 1. Flowchart illustrating the inclusion of patients in the study.**

**Table 1. Demographic and clinical data for the study population that completed the fear of covid 19 questionnaire (responders, n = 10145), compared with patients that did not answer the form, or had incomplete answers (non-responders, n = 7683).**

| | Responders | | Non responders | |
|---|---|---|---|---|
| **Variables** | | | | |
| Gender (n (%)) | | | | |
| Female | 4653 | (45.9) | 3263 | (42.5) |
| Male | 5492 | (54.1) | 4420 | 57.5 |
| Age categories, years (n (%)) | | | | |
| 18–29 | 1756 | (17.3) | 1863 | (24.2) |
| 30–39 | 1749 | (17.2) | 1397 | (18.2) |
| 40–49 | 2071 | (20.4) | 1219 | (15.9) |
| 50–59 | 2331 | (23.0) | 1333 | (17.3) |
| 60–69 | 1532 | (15.1) | 899 | (11.7) |
| 70–79 | 632 | (6.2) | 743 | (9.7) |
| Education (n (%)) | | | | |
| Primary and lower secondary school (10years) | 707 | (7.0) | | |
| Upper secondary school | 3025 | (29.8) | | |
| Vocational school | 1655 | (16.3) | | |
| University college (4 years or less) | 2837 | (28.0) | | |
| University (more than 4 years) | 1863 | (18.4) | | |
| Occupational status (n (%)) | | | | |
| Unemployed (and not under education) | 2700 | (26.6) | | |
| Full-time work | 5476 | (54.0) | | |
| Part-time work | 1081 | (10.7) | | |
| Student | 492 | (4.8) | | |
| Part-time work and student | 298 | (2.9) | | |
| Ethnicity (n (%)) | | | | |
| European | 9223 | (90.9) | 6839 | (89.0) |
| African | 72 | (0.7) | 145 | (1.9) |
| Asian | 110 | (1.1) | 142 | (1.8) |
| Other | 46 | (0.5) | 37 | (0.5) |
| Unknown | 1 | (0.0) | 0 | (0.0) |
| Smoking habits (n (%)) | | | | |
| Never smoker | 5847 | (57.6) | 4437 | (57.8) |
| Daily smoker | 1042 | (10.3) | 927 | (12.1) |
| Ex-smoker | 2839 | (28.0) | 1935 | (25.2) |
| Unknown | 66 | (0.7) | 79 | (1.0) |
| Complications (n (%)) | | | | |
| No complications | 3036 | (29.9) | 2019 | (26.3) |
| Microvascular complications | 3184 | (31.4) | 2310 | (30.1) |
| Macrovascular complications | 1054 | (10.4) | 978 | (12.7) |
| Insulin regime (n (%)) | | | | |
| Insulin pen | 6439 | (63.5) | 5228 | (68.0) |
| Insulin pump | 3563 | (35.1) | 2340 | (30.5) |
| BMI kg/m$^2$ | | | | |
| BMI (mean (SD)) | 26.9 | (4.9) | 26.3 | (5.0) |
| BMI < 25 (n (%)) | 3694 | (36.4) | 3143 | (40.1) |
| BMI 25–30 (n (%)) | 3820 | (37.7) | 2640 | (34.4) |
| BMI > 30 (n (%)) | 2131 | (21.0) | 1469 | (19.1) |
| HbA1c mmol/mol | | | | |
| HbA1c (mean (SD)) | 59.2 | (12.8) | 63.0 | (14.7) |

A high proportion (46.4%) had a college or university degree, and 26.6% were either retired or unemployed.

## Total fear score

The mean total fear score was 13.8 (SD 5.8), and the median score was 12 (range 7–35) (Table 2). "Fig 2" illustrates the mean fear score for each of the seven questions in the questionnaire. As shown, question 1 and question 2 contributed most to the fear of Covid 19, and question 6 and 7 contributed the least.

A total of 11.9% of the patients strongly disagreed with all the questions and had the lowest possible total score on the FCV-19S. For three of the questions, all of which assessed somatic symptoms (q3, q6, and q7), very few patients "agreed" or "strongly agreed" with the question (4.9, 1.6, and 2.8% respectively).

## Fear score assessed against various demographic and clinical variables

**Age and gender.** Women had significantly higher total fear scores than men at 15.1 vs. 12.6 (p<0.05) (Table 2). Women had twice the risk of fear compared with men, with an OR of 1.94 (95% CI 1.79–2.10) (Table 3). "Fig 3A" illustrates the correlation between the fear scores and age in both genders. As shown the total fear score was significantly higher in women than in men (p<0.05), and both genders showed significantly increased fear scores with increasing age (p<0.05). The highest fear score was found in women aged 60–69 years, with a mean total fear score of 15.8 (SD 5.9). There was a significantly higher mean fear score among patients more than 50 years old compared with patients less than 50 years old, 14.2 (5.9) vs. 13.4 (5.7) (p<0.05). We found no significant difference in fear score between those below and above 70 years of age 13.8 (5.8) vs. 13.7 (5.7) (p = 0.75).

**BMI.** "Fig 3B" illustrates a correlation between BMI and total fear score, more pronounced in men with a correlation coefficient of 0.096 compared with 0.062 for women. The differences and correlation were significant (p<0.05).

**Glycemic control, complications, and treatment regimes.** There was a significant correlation between HbA1c and total fear score (rho 0.067, p<0.05), more pronounced for women than men ("Fig 3C"). Patients with HbA1c below 75 mmol/mol had significantly lower fear scores (mean score 13.7, SD 5.76) compared with those with HbA1c above 75 mmol/mol (mean score 14.5, SD 6.36), p<0.05. The difference in mean total fear score between patients treated with multiple daily injection therapy compared with insulin pump was small, but significantly higher in pump users, with a mean score of 13.6 (SD 5.81) and 14.0 (5.83) respectively. There was also significantly higher total fear scores in patients with diabetic micro- and macrovascular complications, compared with patients without complications (Table 2).

**Ethnicity, socioeconomic status and smoking habits.** There was a significantly higher total fear score in non-European patients compared with European patients. Non-Europeans showed a twofold risk of fear compared with Europeans, with an OR of 2.02 (95% CI 1.55–2.63). The mean fear score was 16.7 (SD 7.37) compared with 13.7 (SD 5.77) respectively (p<0.05). The Asian patients showed the highest fear scores with a mean score of 17.8 (SD 7.7). Furthermore, there was a clear significant association between educational status and fear, with a lower fear score with increasing educational level (Table 2). In addition, for unemployed and retired patients the fear score was significantly higher than patients working part time or full time (p<0.05). Daily smokers had significantly higher fear scores than former smokers and non-smokers (p<0.05), and the OR for fear were 1.59 (1.39–1.81) for daily smokers compared with non-smokers.

**Table 2. Showing total fear score in the different patient categories.**

| | Total fear score | |
|---|---|---|
| | **Mean** | **SD** |
| Age | | |
| Age < 50 | 13.4 | 5.7 |
| Age ≥ 50 | 14.2 | 5.9 |
| Age < 70 | 13.8 | 5.8 |
| Age ≥ 70 | 13.7 | 5.7 |
| HbA1c | | |
| HbA1c < 75 | 13.7 | 5.8 |
| HbA1c ≥ 75 | 14.5 | 6.4 |
| Complications | | |
| No complications | 13.6 | 5.7 |
| Microvascular complications | 13.9 | 5.8 |
| Macrovascular complications | 14.6 | 6.1 |
| Insulin device | | |
| Insulin pen | 13.7 | 5.8 |
| Insulin pump | 14.0 | 5.8 |
| Ethnicity | | |
| European | 13.7 | 5.8 |
| Non-european | 16.7 | 7.3 |
| African | 15.5 | 7.2 |
| Asian | 17.8 | 7.7 |
| Other | 16.2 | 6.4 |
| Occupational status | | |
| Unemployed (and not under education) | 15.1 | 6.3 |
| Full-time work | 13.0 | 5.4 |
| Part-time work | 14.9 | 6.1 |
| Student | 12.8 | 5.5 |
| Part-time work and student | 13.0 | 5.1 |
| Education | | |
| Primary and lower secondary school (10years) | 15.9 | 6.9 |
| Upper secondary school | 14.2 | 6.1 |
| Vocational school | 14.1 | 5.9 |
| University college (4 years or less) | 13.2 | 5.4 |
| University (more than 4 years) | 12.7 | 5.2 |
| Smoking habits | | |
| Never smoker | 13.3 | 5.6 |
| Daily smoker | 15.1 | 6.5 |
| Ex-smoker | 14.2 | 5.9 |
| Unknown | 13.8 | 6.3 |

Footnote: For comparison, the mean total fear score across all categories was 13.8 with sd = 5.8.

## Patients that did not answer the questionnaire or answered incompletely

The 7683 patients that did not answer (n = 7611), or answered the questionnaires incompletely (n = 72) had quite similar demographic and clinical data when compared to patients who answered all questions. See Table 1 for comparison.

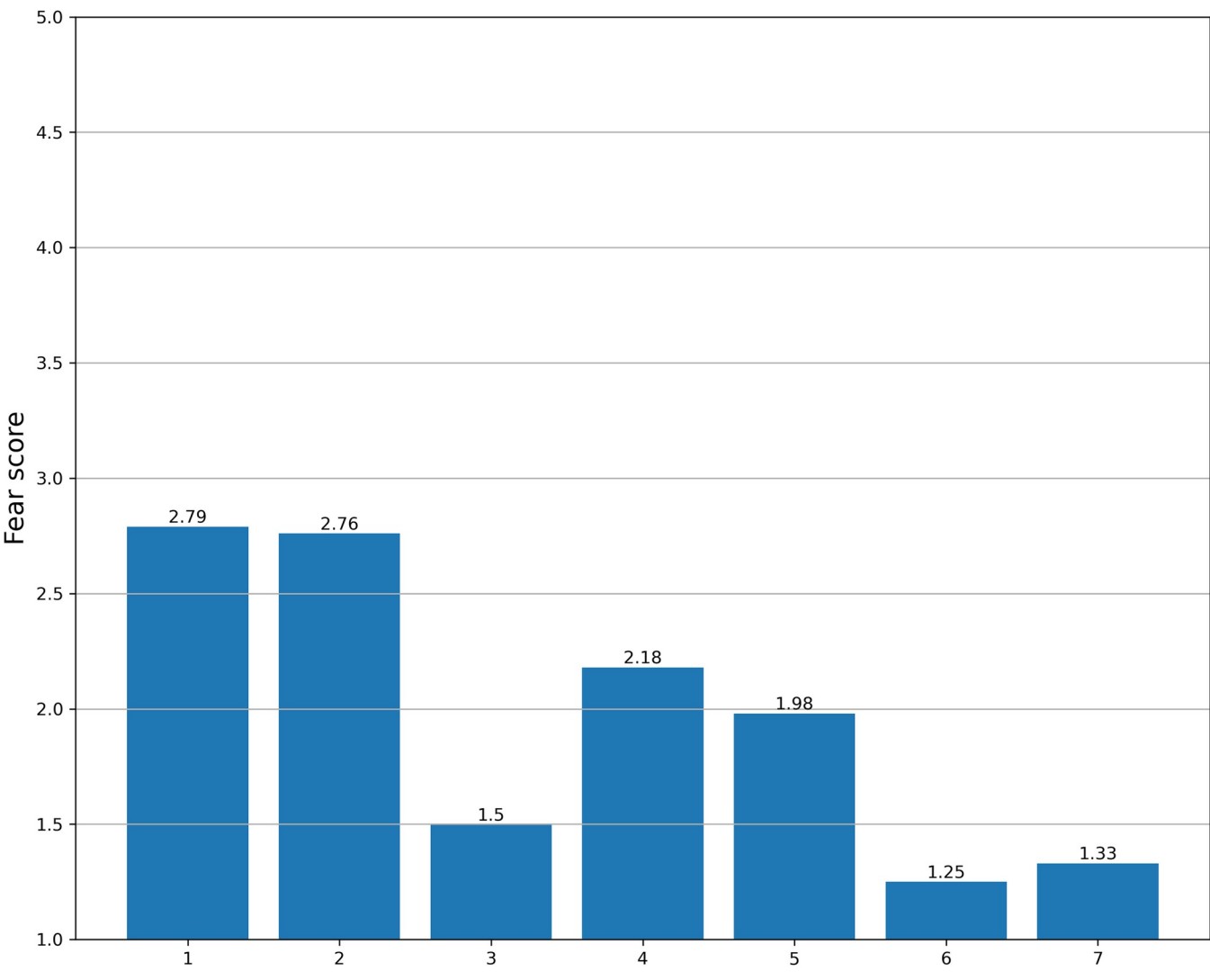

**Fig 2. Mean score of fear for the seven questions in the fear of Covid 19 scale in the whole study population.**

## Discussion

To the best of our knowledge, the present study comprises the largest cohort of patients assessed with a validated fear of Covid 19 scale reported to date. Approximately 52% (n = 10217) of the Norwegian adult type 1 diabetes population were included in the study [16]. The results showed an overall low level of fear of Covid 19. The fear score increased with increasing age and was more pronounced in women than men. Our data also showed a significant correlation between impaired glycemic control and fear, and an association between fear scores and the presence of diabetic complications.

The total fear score was generally low, with a total mean score of 13.8/35 for the whole diabetes population. This could be because the questionnaire was sent out in the end of the third wave of infection in Norway (May 2021), when patients had been exposed to the pandemic for 15 months. In addition, by May 2021, 32% of the adult population had received one vaccine dose, and 8.6% had received two doses [17]. Our findings are supported by a study from Denmark by Musche et al [18], which studied anxiety and fear of Covid 19 among diabetic patients

**Table 3.** Table showing odds ratio (OR) of fear for different variables of interest.

| Characteristics | Univariate analysis | | | Multivariate analysis | | |
|---|---|---|---|---|---|---|
| | OR | 95% CI | P-value | OR | 95% CI | P-value |
| **Gender** | | | | | | |
| Male | ref | | | ref | | |
| Female | 1.94 | 1.79–2.10 | <0.001 | 1.98 | 1.82–2.16 | <0.001 |
| **European** | | | | | | |
| Yes | ref | | | Ref | | |
| No | 2.02 | 1.55–2.63 | <0.001 | 1.98 | 1.50–2.61 | <0.001 |
| **Smoking** | | | | | | |
| Never smoker | ref | | | ref | | |
| Daily smoker | 1.59 | 1.39–1.81 | <0.001 | 1.44 | 1.24–1.66 | <0.001 |
| Ex-smoker | 1.30 | 1.19–1.42 | <0.001 | 1.31 | 1.18–1.44 | <0.001 |
| Unknown | 1.25 | 0.77–2.05 | 0.365 | 1.13 | 0.67–1.91 | 0.650 |
| **Unemployed** | | | | | | |
| No | ref | | | | | |
| Yes | 1.80 | 1.65–1.97 | <0.001 | 1.7 | 1.54–1.87 | <0.001 |

earlier in the pandemic (April 2020). They found that generalized anxiety and depression were similar in patients with diabetes and healthy controls, but the diabetic patients reported higher Covid 19-related fear, increased risk perception, and behavioral changes.

As illustrated in "Fig 3A", the fear score was higher in women than men. Our findings are in line with findings from Basit et al [19] who assessed the fear of Covid 19 among a subgroup of Pakistani patients with type 2 diabetes. However, this result is in contrast to the finding of Ahorsu et al, who has developed the fear of Covid 19 scale in the general Iranian population. They found that gender had no effect on the fear score [14].

"Fig 3A" shows a significant positive correlation between age and fear scores, stronger for men than for women. The finding is reasonable as older age was launched as a risk factor for serious disease and bad outcome early in the pandemic. The finding is however in conflict with findings from Basit et al, who found no association between fear scores and age [19], and findings from Lee et al, who assessed the coronavirus anxiety in healthy Turkish people. They found that younger people were more afraid of Covid 19 than older people [20].

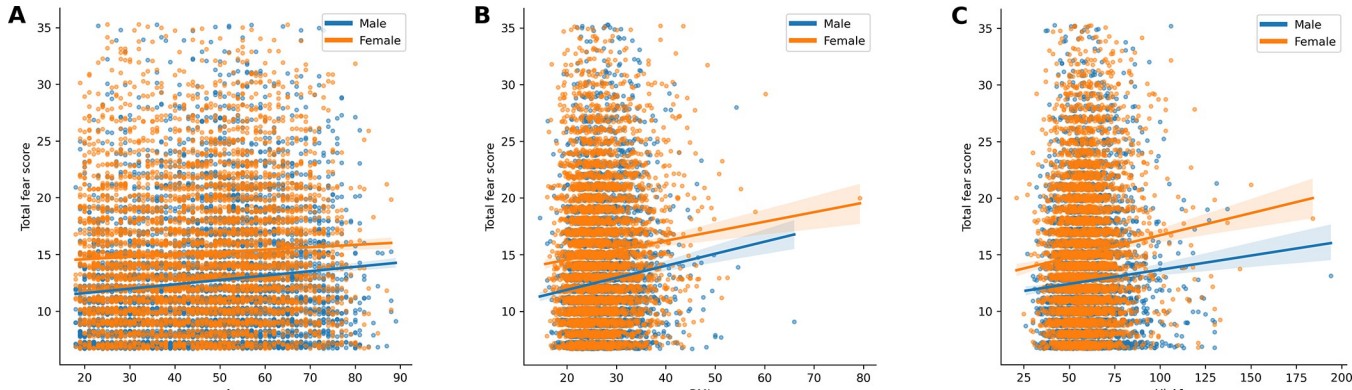

**Fig 3.** Correlation between fear score and increasing age (A), BMI (B) and HbA1c (C) for women and men. The difference and correlations are statistically significant with p<0.05.

Early in the pandemic obesity was highlighted as a risk factor for serious disease and increased mortality [21, 22]. We found higher BMI to be associated with increased fear scores ("Fig 3B"), and more pronounced for men than for women. In contrast, Kizilkaya et al [23] showed in their study of 568 obese Turkish patients that the population with the highest BMI (above 50 kg/m2) showed less fear of Covid 19, accessed by FCV-19S. However, these patients had a considerably higher BMI than the obese patients in our cohort and are probably not directly comparable.

Early in the pandemic, diabetes mellitus was flagged as an independent risk factor for poor prognosis and fatal outcome in Covid 19 infections. This was later adjusted to be true for type 2 diabetes and obesity, more than diabetes mellitus in general. We found fear scores to be correlated to HbA1c-level. There were also a significantly higher fear score in the group of patients with diabetic complications compared with patients without complications, with the highest fear score in patients with macrovascular complications. Serin et al found the same pattern for patients with diabetes type 2 in Turkey [24]. To the best of our knowledge, there are no other studies comparing the fear of Covid 19 scores to the glycemic control, treatment and complications in type 1 diabetic patients. Although a large Danish study has mapped Covid 19-specific worries and overall psychosocial health among people with diabetes in the initial phase of the Covid 19 pandemic in Denmark [25]. Their results showed that being female, having type 1 diabetes, diabetes complications and diabetes distress, feeling isolated and lonely, and having changed diabetes behaviors were associated with being more worried about Covid 19.

Non-European patients showed significantly higher fear scores compared with patients of European descent. Of the non-European patients, Asian patients had the highest fear scores. In addition, we found that patients with a higher level of education experienced less fear than patients with a lower level of education. The study of Lee et al revealed opposite findings, higher fear with higher educational level.

The World Health Organization released early in the pandemic a statement warning that smokers are more likely to experience a severe Covid 19 illness relative to non-smokers [26], a warning broadcasted in media outlets [27] and supported by scientific studies [28, 29]. Our findings revealed that smokers had significantly greater fear than former smokers and non-smokers. Basit et al showed similar findings with an odds ratio as high as four between non-smokers and daily smokers in the Pakistani type 2 diabetic population [19].

A large study performed in Bergen, Norway in June 2020 assessing fear of Covid 19 in the background population found similar finding to ours [15]. A higher FCV-19S score was positively associated with being female, older age groups, and lower socioeconomic status (lower education and income).

The high number of patients and the high response rate among type 1 diabetic patients are strengths of this study. The studied population is thought to be representative of type 1 diabetes patients in Norway. The results of the study should be interpreted in light of the prevalence of Covid 19 at the time of the data collection, and a possible limitation of our study is the relatively low level of viral transmission in the society when the questionnaire was distributed. In addition, the digital collection of data could have introduced bias into our results. The large number of participants could make the study "over powered", finding statistically significant results that are rather weak and not clinically relevant. Nevertheless, our findings match quite well with what is expected based on patient groups flagged as high risk individuals by the health authorities and should help clinicians to identify subgroups of diabetes patients that may require addition psychological support to address fear of Covid 19 during diabetes follow-up.

## Conclusion

In conclusion, we present a valid assessment of the fear of Covid 19 status in the population of patients with type 1 diabetes in Norway. Overall, we found that the fear of Covid 19 was low during the third wave of infection. Patients considered being at high risk of serious disease by the authorities, such as older individuals, smokers and obese individuals showed a higher level of fear than low risk individuals did. In addition, women, non-European individuals, patients with a lower level of education and not in regular employment showed a higher degree of fear. Finally, we revealed a higher degree of anxiety for Covid 19 in patients with poor glycemic control and diabetes patients with vascular complications. The need for additional psychological support among patient groups found to have increased fear of Covid 19 should be assessed during the diabetic consultation. Finally, the methodological principle with digital collection of data directly from the patient is a novel and exciting way to enrich clinical studies in the future.

## Acknowledgments

All authors report no conflict of interest regarding this study. All authors contributed to the study, its concept and design, acquisition of data, and analysis and interpretation of data. G.Å. U. drafted the manuscript. All authors provided critical revision of the manuscript for important intellectual content. T.E. provided statistical analysis. G.Å.U. is the guarantor of this work and, as such, had full access to all of the data in the study and takes responsibility for the integrity of the data and the accuracy of the data analysis.

## Author Contributions

**Data curation:** Grethe Åstrøm Ueland, John Graham Cooper.

**Formal analysis:** Grethe Åstrøm Ueland, Tony Ernes, Tone Vonheim Madsen, Eystein Sverre Husebye, Sverre Sandberg, Karianne Fjell Løvaas, John Graham Cooper.

**Investigation:** Grethe Åstrøm Ueland, Tone Vonheim Madsen, Eystein Sverre Husebye, Sverre Sandberg, Karianne Fjell Løvaas, John Graham Cooper.

**Methodology:** Grethe Åstrøm Ueland, Tony Ernes, Tone Vonheim Madsen, Eystein Sverre Husebye, Sverre Sandberg, Karianne Fjell Løvaas, John Graham Cooper.

**Project administration:** Grethe Åstrøm Ueland, Tone Vonheim Madsen, Sverre Sandberg, Karianne Fjell Løvaas, John Graham Cooper.

**Resources:** Tone Vonheim Madsen, Sverre Sandberg, Karianne Fjell Løvaas, John Graham Cooper.

**Software:** Grethe Åstrøm Ueland, Tony Ernes, Tone Vonheim Madsen, Eystein Sverre Husebye, Sverre Sandberg, Karianne Fjell Løvaas, John Graham Cooper.

**Supervision:** Grethe Åstrøm Ueland, Eystein Sverre Husebye, Sverre Sandberg, Karianne Fjell Løvaas, John Graham Cooper.

**Validation:** Grethe Åstrøm Ueland, Tony Ernes, Tone Vonheim Madsen, Sverre Sandberg, Karianne Fjell Løvaas, John Graham Cooper.

**Visualization:** Grethe Åstrøm Ueland, Tony Ernes, Tone Vonheim Madsen, Karianne Fjell Løvaas, John Graham Cooper.

**Writing – original draft:** Grethe Åstrøm Ueland, Tony Ernes, Tone Vonheim Madsen, Eystein Sverre Husebye, Sverre Sandberg, Karianne Fjell Løvaas, John Graham Cooper.

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
