## [Decision Letter · Decision Letter 0]

28 Mar 2022

PONE-D-22-07616Fear of Covid 19 during the third wave of infection in Norwegian patients with type 1 diabetesPLOS ONE

Dear Dr. Ueland,

Thank you for submitting your manuscript to PLOS ONE. After careful consideration, we feel that it has merit but does not fully meet PLOS ONE’s publication criteria as it currently stands. Therefore, we invite you to submit a revised version of the manuscript that addresses the points raised during the review process.

I have received the reports from our advisors on your manuscript which you submitted to PLOS ONE.

Based on the comments received, I feel that your manuscript could be reconsidered for publication should you be prepared to incorporate major revisions.

When preparing your revised manuscript, you are asked to carefully consider the reviewer comments below and submit a list of responses to the comments.

Editor Comments: The paper should be checked by a professional speaker of English before complete acceptance.

We look forward to receiving your revised manuscript.

Kind regards,

Muhammad Sajid Hamid Akash

Academic Editor

PLOS ONE

Journal Requirements:

4. Please include a copy of Table 2 which you refer to in your text on page 7.

5. Please include captions for your Supporting Information files at the end of your manuscript, and update any in-text citations to match accordingly. Please see our Supporting Information guidelines for more information: http://journals.plos.org/plosone/s/supporting-information

Reviewers' comments:

Reviewer's Responses to Questions

**Comments to the Author**

1. Is the manuscript technically sound, and do the data support the conclusions?

Reviewer #1: Partly

2. Has the statistical analysis been performed appropriately and rigorously? 

Reviewer #1: No

3. Have the authors made all data underlying the findings in their manuscript fully available?

Reviewer #1: No

4. Is the manuscript presented in an intelligible fashion and written in standard English?

Reviewer #1: No

5. Review Comments to the Author

Reviewer #1: 1. Grammatical, formatting and spacing mistakes should be corrected in the manuscript. Font color should be black.

2. Introduction should include background of COVID-19 and type 1 diabetes especially prevalence, symptoms of COVID-19, its main variants and the role of immunity in prevention of this viral disease. Following articles can be used as references:

(Endocr Metab Immune Disord Drug Targets. 2022; https://doi.org/10.2174/1871530322666220110113028).

(J Med Virol. 2021; https://doi.org/10.1002/jmv.27256).

3. Reference format should be modified according to journal.

4. Most of the participants were male (54.1%) in your study but fear factor is more in female then man. justify this statement.

5. Result is too long, try to trim it by removing extra explanation.

6. Be specific with your topic and shorten your discussion up to 2 pages. References are also missing in most of the sentences in discussion.

7. What are the limitations of your study?

8. What is the future perspective of your study?

6. PLOS authors have the option to publish the peer review history of their article (what does this mean?). If published, this will include your full peer review and any attached files.

Reviewer #1: **Yes: **Sibgha Noureen

---

## [Author Response · Author response to Decision Letter 0]

7 Apr 2022

Point-to-point rebuttal letter

Thank you for the constructive comments on our manuscript PONE-D-22-07616

“Fear of Covid 19 during the third wave of infection in Norwegian patients with type 1 diabetes”

 We have now carefully revised the paper according to the comments by the reviewer as detailed below. Changes in the manuscript are made in red. We hope you will find the revised version acceptable for publication.

Editor Comments: The paper should be checked by a professional speaker of English before complete acceptance.

The paper has now been checked by last author of the manuscript, John Cooper, who is a native Englishman. 

Journal Requirements:

The manuscript is now formatted according to PLOS Ones style requirements. 

All the participants have given written informed consent. This information is now included in the method section, page 7 line 142. 

Sharing of individual participant data with third parties was not specifically included in the informed consent of the study, and unrestricted distribution of such data may pose a potential threat of revealing participants’ identities. To minimise this risk, researchers who wish to inquire about access to individual participant data that underlie the results reported in this article can submit a request to the The Norwegian Adult Diabetes registry (noklus @noklus.no). To gain access, researchers will need to sign a data access agreement and obtain the approval of the local ethics committee.

4. Please include a copy of Table 2 which you refer to in your text on page 7.

We apologize for the mistake, as Table 2 was identical with Supplementary Table S1. We have now chosen to include Supplementary Table S2 in the main manuscript, as Table 3. We no longer need a supplementary file. The alterations in the numbering of tables are now marked in red throughout the manuscript. 

5. Please include captions for your Supporting Information files at the end of your manuscript, and update any in-text citations to match accordingly. Please see our Supporting Information guidelines for more information: http://journals.plos.org/plosone/s/supporting-information

Supporting/supplementary file is no longer included in the manuscript. 

Reviewers' comments:

Reviewer's Responses to Questions

Comments to the Author

1. Is the manuscript technically sound, and do the data support the conclusions?

Reviewer #1: Partly

2. Has the statistical analysis been performed appropriately and rigorously? 

Reviewer #1: No

3. Have the authors made all data underlying the findings in their manuscript fully available?

Reviewer #1: No

4. Is the manuscript presented in an intelligible fashion and written in standard English?

Reviewer #1: No

5. Review Comments to the Author

Reviewer #1: 1. Grammatical, formatting and spacing mistakes should be corrected in the manuscript. Font color should be black.

We have reviewed the manuscript carefully, and hope that all grammatical, formatting and spacing mistakes are corrected. Font color is black. 

2. Introduction should include background of COVID-19 and type 1 diabetes especially prevalence, symptoms of COVID-19, its main variants and the role of immunity in prevention of this viral disease. Following articles can be used as references:

(Endocr Metab Immune Disord Drug Targets. 2022; https://doi.org/10.2174/1871530322666220110113028).

(J Med Virol. 2021; https://doi.org/10.1002/jmv.27256).

Thank you for this valuable suggestion of improvement, we have now included this in the introduction page 3, line 47-53 and page 4 line 73-74.

3. Reference format should be modified according to journal.

We have now modified the reference to style Vancouver as required.

4. Most of the participants were male (54.1%) in your study but fear factor is more in female then man. justify this statement.

You are correct, there are more men than women with type 1 diabetes in Norway, and our study population is therefore representative for the actual patient population. In addition, we have a high number of study participants, and the statistical power is high- so the fact that more men than women are participating is not affecting the results. 

5. Result is too long, try to trim it by removing extra explanation.

Thank you- we have now trimmed the results section from the original 854 words to 734 words. 

6. Be specific with your topic and shorten your discussion up to 2 pages. References are also missing in most of the sentences in discussion.

We agree, the discussion was too long. We have shortened it from 1862 words to 1445 words, which corresponds to 4 pages instead of 6 (discussion without conclusion).

7. What are the limitations of your study?

We agree that the limitations of the study should have been presented more explicitly. We have addressed this problem with a revised section on page 14, line 315-322:

The results of the study should be interpreted in light of the prevalence of COVID 19 in the country at the time of the data collection. In Norway, the first cases of COVID 19 were confirmed in February 2020. The peak of the outbreak was during March and April 2020. The third wave of infection in Norway started in the middle of March 2021, and as of the 1st of May 2021 (when the fear questionnaire was sent out), the total number of confirmed Covid 19 cases was 113899 and there had been 766 deaths from Covid 19 (31). The relatively low level of viral transmission in the society when the questionnaire was distributed, and the digital collection of data could have biased our results, and is an important limitation of the study.

The large number of participants could make the study “over powered”, finding statistic significances that are rather weak and not clinically relevant.

8. What is the future perspective of your study?

Regarding future perspective of the study, we have presented the findings of this study at a national meeting for both doctors and nurses- to encourage vigilance in this area at diabetic outpatient clinics and, if necessary, to provide psychological help to those with a high level of fear of Covid 19.

Furthermore we have distributed a much larger questionnaire to all diabetes patients (both DM1 and DM2) in Norway registered in the national register for adults to assess how these patients have coped with the pandemic overall. The results are to be processed. 

In addition, we think that the methodological principle with digital collection of data directly from the patient is a novel and exciting way to enrich clinical studies in the future. (Included in the conclusion of the manuscript)

---

## [Decision Letter · Decision Letter 1]

30 May 2022

PONE-D-22-07616R1Fear of Covid 19 during the third wave of infection in Norwegian patients with type 1 diabetesPLOS ONE

Dear Dr. Ueland,

Thank you for submitting your manuscript to PLOS ONE. After careful consideration, we feel that it has merit but does not fully meet PLOS ONE’s publication criteria as it currently stands. Therefore, we invite you to submit a revised version of the manuscript that addresses the points raised during the review process.

I have received the reports from our advisors on your manuscript which you submitted to PLOS ONE.

Based on the comments received, I feel that your manuscript could be reconsidered for publication should you be prepared to incorporate major revisions.

When preparing your revised manuscript, you are asked to carefully consider the reviewer comments below and submit a list of responses to the comments.

Editor Comments: There is a huge list of grammatical mistakes and syntax errors. The paper should be checked by a professional speaker of English before complete acceptance.

We look forward to receiving your revised manuscript.

Kind regards,

Muhammad Sajid Hamid Akash

Academic Editor

PLOS ONE

Reviewers' comments:

Reviewer's Responses to Questions

**Comments to the Author**

1. If the authors have adequately addressed your comments raised in a previous round of review and you feel that this manuscript is now acceptable for publication, you may indicate that here to bypass the “Comments to the Author” section, enter your conflict of interest statement in the “Confidential to Editor” section, and submit your "Accept" recommendation.

Reviewer #1: (No Response)

2. Is the manuscript technically sound, and do the data support the conclusions?

Reviewer #1: No

3. Has the statistical analysis been performed appropriately and rigorously? 

Reviewer #1: Yes

4. Have the authors made all data underlying the findings in their manuscript fully available?

Reviewer #1: Yes

5. Is the manuscript presented in an intelligible fashion and written in standard English?

Reviewer #1: No

6. Review Comments to the Author

Reviewer #1: 1. Summarize the whole manuscript comprehensively in the abstract (limit the words up to 250).

2. Figure 3 is not clear, re-draw this figure and add caption of graphs as well.

3. As you mentioned that you have cite the suggested paper on page 4 line 73-74, but in your manuscript suggested paper is cited on line 72 with reference # 8 rather than reference # 9. It means the sequence of references is incorrect.?

4. Font color should be black in introduction. Re-check your font setting and correct it.

5. Paragraph setting of whole manuscript is required as well. Paragraph setting should be justified.

6. Try to trim the discussion furthermore (maximum 3 pages).

7. Conclude your findings under a separate heading.

7. PLOS authors have the option to publish the peer review history of their article (what does this mean?). If published, this will include your full peer review and any attached files.

Reviewer #1: **Yes: **Sibgha Noureen

---

## [Author Response · Author response to Decision Letter 1]

8 Jun 2022

Point-to-point rebuttal letter

Thank you for the constructive comments on our manuscript PONE-D-22-07616

“Fear of Covid 19 during the third wave of infection in Norwegian patients with type 1 diabetes”

 We have addressed the comments of the reviewers in the revised paper. Changes in the manuscript are highlighted in red. The manuscript has been checked for grammar and syntax by native English speaker. We hope you will find the revised version acceptable for publication.

Reviewer #1:

1. Summarize the whole manuscript comprehensively in the abstract (limit the words up to 250).

We think the abstract comprehensively covers the whole manuscript, and it consists of 240 words. 

2. Figure 3 is not clear, re-draw this figure and add caption of graphs as well.

Figure 3 is now re-drawn, and section 3A, 3B and 3C are submitted as individual figures to secure better resolution. All graphs have captions. The figure legend is also updated/improved. 

3. As you mentioned that you have cite the suggested paper on page 4 line 73-74, but in your manuscript suggested paper is cited on line 72 with reference # 8 rather than reference # 9. It means the sequence of references is incorrect.?

The sequences of references are correct, however the page and line numbers were incorrect. We have rectified the error.

Suggested reference (Endocr Metab Immune Disord Drug Targets. 2022; https://doi.org/10.2174/1871530322666220110113028) is reference 1, page 3 line 52. 

And suggested reference (J Med Virol. 2021; https://doi.org/10.1002/jmv.27256) is reference 8 at page 4 line 72.

4. Font color should be black in introduction. Re-check your font setting and correct it.

Font color is black in the introduction.

5. Paragraph setting of whole manuscript is required as well. Paragraph setting should be justified.

We have reassessed the paragraph setting of whole manuscript, and justified it.

6. Try to trim the discussion furthermore (maximum 3 pages).

We have shortened the discussion substantially, and it is now just over 3 pages. 

7. Conclude your findings under a separate heading.

We have now established a separate heading for Conclusion (line 298).

---

## [Editor Report · Decision Letter 2]

20 Jun 2022

PONE-D-22-07616R2

Fear of Covid 19 during the third wave of infection in Norwegian patients with type 1 diabetes

PLOS ONE

Dear Dr. Ueland,

Thank you for submitting your manuscript to PLOS ONE. After careful consideration, we have decided that your manuscript does not meet our criteria for publication and must therefore be rejected.

The authors have not addressed all the comments raised by the reviewer. The responses are not sufficient to re-review this paper. My decision is to reject this paper at this stage.

I am sorry that we cannot be more positive on this occasion, but hope that you appreciate the reasons for this decision.

Kind regards,

Muhammad Sajid Hamid Akash

Academic Editor

PLOS ONE

- - - - -

---

## [Author Response · Author response to Decision Letter 2]

1 Jul 2022

Point-to-point rebuttal letter

Thank you for the constructive comments on our manuscript PONE-D-22-07616

“Fear of Covid 19 during the third wave of infection in Norwegian patients with type 1 diabetes”

 We have addressed the comments of the reviewers in the revised paper. Changes in the manuscript are highlighted in red. The manuscript has been checked for grammar and syntax by native English speaker. We hope you will find the revised version acceptable for publication.

Reviewer #1:

1. Summarize the whole manuscript comprehensively in the abstract (limit the words up to 250).

We think the abstract now comprehensively covers the whole manuscript, and it consists of 240 words. 

2. Figure 3 is not clear, re-draw this figure and add caption of graphs as well.

Figure 3 is now re-drawn, and section 3A, 3B and 3C are submitted as individual figures to secure better resolution. All graphs have captions. The figure legend is also updated/improved. 

If you want the figure to look different, please specify, and we will try to change it further.

3. As you mentioned that you have cite the suggested paper on page 4 line 73-74, but in your manuscript suggested paper is cited on line 72 with reference # 8 rather than reference # 9. It means the sequence of references is incorrect.?

The sequences of references are correct, however the page and line numbers were incorrect. We have rectified the error.

Suggested reference (Endocr Metab Immune Disord Drug Targets. 2022; (https://doi.org/10.2174/1871530322666220110113028) is referred in reference 1, page 3, line 52. 

And suggested reference (J Med Virol. 2021; https://doi.org/10.1002/jmv.27256) is reference 8 at page 4 line 72.

4. Font color should be black in introduction. Re-check your font setting and correct it.

Font color is changed to black in the introduction, and the whole manuscript.

5. Paragraph setting of whole manuscript is required as well. Paragraph setting should be justified.

We have reassessed the paragraph setting of whole manuscript.

The introduction is divided into paragraphs without subheadings. 

The method section is subdivided into paragraphs with the following subheadings: Participants/data collection, Measurements, Fear of Covid 19 scale, Statistical analysis and Ethical considerations. 

The result section is divided into paragraphs with the following headings:

Study population, Fear score assessed against various demographic and clinical variables, patients that did not answer or answered incomplete. 

The discussion part is divided into paragraphs without subheadings.

6. Try to trim the discussion furthermore (maximum 3 pages).

We have shortened the discussion substantially from 1632 words to 1047 words, and it is now 3 pages. 

7. Conclude your findings under a separate heading.

We have now established a separate heading for Conclusion (line 299).

---

## [Editor Report · Decision Letter 3]

13 Jul 2022

Fear of Covid 19 during the third wave of infection in Norwegian patients with type 1 diabetes

PONE-D-22-07616R3

Dear Dr. Ueland,

We’re pleased to inform you that your manuscript has been judged scientifically suitable for publication and will be formally accepted for publication once it meets all outstanding technical requirements. Your response to prior reviews has resulted in a large improvement in the manuscript. 

Kind regards,

Edward Jay Trapido, ScD

Academic Editor

PLOS ONE

Additional Editor Comments (optional):

Thank you for being responsive to the reviews and for your multiple submissions. I believe your comments and changes have been useful.
---

## [Editor Report · Acceptance letter]

18 Jul 2022

PONE-D-22-07616R3 

Fear of Covid 19 during the third wave of infection in Norwegian patients with type 1 diabetes 

Dear Dr. Ueland:

I'm pleased to inform you that your manuscript has been deemed suitable for publication in PLOS ONE. Congratulations! Your manuscript is now with our production department. 

Kind regards, 

on behalf of

Dr. Edward Jay Trapido 

Academic Editor

PLOS ONE